# Consistency Matters: Neural ODE Parameters are Dependent on the Training Numerical Method

**C. Coelho [1], M. Fernanda P. Costa [1], L.L. Ferrás [1,2]**
[1]Centre of Mathematics (CMAT), University of Minho
[2]Department of Mechanical Engineering (Section of Mathematics), FEUP - University of Porto
`cmartins@cmat.uminho.pt, mfc@math.uminho.pt, lferras@fe.up.pt`

## Abstract

Neural Ordinary Differential Equations (Neural ODEs) are continuous-depth models that use an ordinary differential equation (ODE) to capture the dynamics of data. Due to their modelling capabilities several works on applications and novel architectures using Neural ODEs can be found in the literature. In this work, we call for the attention to the need of using the same numerical method for both training and making predictions with Neural ODEs since the numerical method employed influences the prediction process, thereby impacting the loss function and introducing variance into parameter optimisation. We provide theoretical insights into how numerical methods of varying orders or with different step sizes influence the loss function of the network. To validate our theoretical analysis, we conduct a series of simple preliminary numerical experiments employing a regression task, demonstrating how the training numerical method influences model performance for testing. Our findings underscore the need for consistency in numerical methods for training and prediction, a consideration not previously emphasised or documented in the literature.

## 1 Introduction

Traditional Neural Networks (NNs) typically consist of a discrete sequence of layers tasked with mapping input data to corresponding outputs. However, they face limitations in handling time-dependent systems effectively and are restricted to making one prediction per input.

Neural Ordinary Differential Equations (Neural ODEs) are continuous-depth models that use an ODE to capture the dynamics of data. Unlike traditional NNs, Neural ODEs inherently account for the time-dependence of inputs, enabling predictions at arbitrary time points (Chen et al., 2018; Krishnapriyan et al., 2023; Onken & Ruthotto, 2020). This distinctive capability has attracted significant attention, leading to various applications in real-world problems, as evidenced by the growing literature (Portwood et al., 2019; Xing et al., 2022; Bansude et al., 2023). Furthermore, researchers have proposed various architectures that incorporate Neural ODEs to enhance modelling capabilities, capitalising on their flexibility and expressive power to address various tasks (Rubanova et al., 2019; Yildiz et al., 2019; Khoshsirat & Kambhamettu, 2023).

An integral aspect of Neural ODEs is the incorporation of a numerical method to solve the ODE being learnt. Numerous choices of numerical methods exist in the literature, with selecting the appropriate one posing a challenge due to the need to balance accuracy and computational cost. Moreover, besides being essential for training, a numerical method is also indispensable for making predictions using the learnt ODE. Interestingly, while the literature provides guidance on selecting numerical methods for solving ODEs (Chapra, 2010), there is a notable absence of explicit direction regarding whether the same numerical method should be used for both training and making predictions in the context of Neural ODEs.

In numerical methods theory, it is well-established that in the limit where the step size approaches infinitesimal values, any numerical method, irrespective of its approximation error and computational scheme, will compute the same solution. However, in the realm of Neural ODEs, the numerical

method employed influences the prediction process, thereby impacting the loss function and introducing variance into parameter optimisation. This variance introduced by the choice of numerical method during training can affect the optimisation of model parameters, potentially influencing convergence behaviour and overall model performance.

In this work, we underscore the critical importance of thoughtful consideration when selecting a numerical method for making predictions with Neural ODEs. We provide theoretical insights into how numerical methods of varying orders or with different step sizes influence the loss function of Neural ODEs, thereby demonstrating that the choice of numerical method inherently impacts the parameters of the network. To complement our statements, we present numerical experiments employing a simple regression task. Through this experiments, we showcase the performance disparities exhibited by Neural ODEs when trained and tested using the same versus different numerical methods. Ultimately, our aim is to raise awareness within the research and industry community about the significance of employing consistent numerical methods for both training and prediction in Neural ODEs and, by extension, in any architecture that incorporates a numerical method.

This paper is organised as follows. In Section 2, we provide a concise review of Neural ODEs and numerical methods, giving the essential background needed for understanding the subsequent discussions. Section 3 delves into our theoretical analysis, demonstrating how the order and step size of the numerical method impact the loss function of Neural ODEs. Additionally, we present numerical results corroborating these theoretical findings. Finally, we conclude our work in Section 4 by summarising the key findings.

## 2 BACKGROUND

In this section, we give a briefly highlight of the fundamentals of Neural ODEs and a short introduction to numerical methods, elucidating their impact on the solution of ODEs.

### 2.1 NEURAL ODES

Chen et al. (2018) introduced Neural ODEs, continuous-depth models that define the output as the solution of an Ordinary Differential Equation (ODE). Thus, Neural ODEs learn to model the underlying dynamics of data by fitting an ODE. To do this, Neural ODEs are composed of two main components: a NN with parameters $\boldsymbol{\theta}$, $\boldsymbol{f_\theta}$, that models the right-hand side of an ODE; and a numerical method, $ODESolve$, that solves the ODE and outputs the predictions:

$$\hat{\boldsymbol{y}}_m(\boldsymbol{\theta})_{m=1,\ldots,M} = ODESolve(\boldsymbol{f_\theta}, \boldsymbol{y}_0, (t_0, t_M)), \ \text{ with } \ m = 1, \ldots, M, \tag{1}$$

where $\hat{\boldsymbol{y}}_m(\boldsymbol{\theta})$ is the prediction at time step $t_m$ with $m = 1, \ldots, M$ and $M$ the number of data points. The numerical method solves the ODE as an initial value problem with initial condition $(\boldsymbol{y_0}, t_0)$, *i.e.* the data point at the first time step, in the time interval $(t_0, t_M)$.

Neural ODEs are trained similarly to traditional NNs, by minimising a loss function $l(\boldsymbol{\theta})$. When training finishes, to make predictions the resulting ODE, $\dfrac{d\hat{\boldsymbol{y}}}{dt} = \boldsymbol{f_\theta}$, must be solved using a numerical method (Chen et al., 2018).

### 2.2 NUMERICAL METHODS

Numerical methods play a crucial role in solving differential equations when analytical solutions are either impossible or intractable, which is often the case in real-world applications. These methods encompass a diverse array of techniques designed to address specific types of differential equations or to optimize computational efficiency while maintaining solution accuracy (Chapra, 2010).

Fundamentally, numerical methods discretize the continuous differential equations into discrete time steps with size $h$, iteratively approximating the solution at the desired time step. There are two distinct approaches to the discretization: fixed-step methods discretize the solution domain at predetermined fixed intervals $h$; adaptive-step methods dynamically adjust the step size based on the local solution behaviour by refining the discretization where the solution evolves rapidly and coarsening

where the solution changes slowly. Additionally, numerical methods employ varying computation schemes and have different orders, *i.e.* rate at which the error decreases as the step size decreases, that offer higher precision or lower computational cost (Chapra, 2010).

The official Neural ODE library, *Torchdiffeq*, offers several choices of fixed-step and adaptive-step methods for solving ODEs. The choices available with their corresponding error are as follows (Chen, 2018):

- **Fixed-step methods:** Euler, first-order method $\mathcal{O}(h^1)$; Midpoint, Second-order method $\mathcal{O}(h^2)$; Fourth-order Runge-Kutta with 3/8 rule, $\mathcal{O}(h^4)$, denoted rk4; Explicit Adams-Bashforth, fourth-order is used $\mathcal{O}(h^4)$, denoted explicit_adams; Implicit Adams-Bashforth-Moulton, fourth-order is used $\mathcal{O}(h^4)$, denoted implicit_adams.

- **Adaptive-step methods:** Runge-Kutta of order 8 of Dormand-Prince-Shampine, $\mathcal{O}(h^8)$, denoted dopri8; Runge-Kutta of order 5 of Dormand-Prince-Shampine, $\mathcal{O}(h^5)$, denoted dopri5; Runge-Kutta of order 3 of Bogacki-Shampine, $\mathcal{O}(h^3)$, denoted bosh3; Runge-Kutta-Fehlberg of order 2, $\mathcal{O}(h^2)$, denoted fehlberg2; Runge-Kutta of order 2, $\mathcal{O}(h^2)$, , denoted adaptive_heun.

Understanding how the order of a numerical method influences on the solution of an ODE is crucial for selecting an appropriate method based on the computational cost versus accuracy trade-off. Higher-order methods offer higher precision but at increased computational cost. Conversely, lower-order methods are computationally cheaper but may produce less accurate solutions, especially in systems with complex dynamics or rapid changes (Chapra, 2010).

## 3    THE DEPENDENCE ON THE NUMERICAL METHOD

In this section, we demonstrate and analyse how employing different numerical methods, distinct from the one used for training, impacts the performance of making predictions with Neural ODEs. First, we provide a theoretical exposition of the contribution of the numerical method to the training and testing of a Neural ODE, followed by empirical evidence from experimental results.

### 3.1   THEORETICAL EXPOSITION

Numerical methods inherently introduce errors during the approximation of ODE solutions. This error is described in terms of the step size $h$ and the order of the method $p$, denoted as $E_{\text{local}}(h, p) = \mathcal{O}(h^p)$. Then, the global error over $N$ integration steps for each prediction point $m$ is denoted by,

$$E_{\text{global}}^m(h, p) = \sum_{n=1}^{N} E_{\text{local}}^n(h, p).$$

As the number of steps $N$ increases, the accumulated error grows linearly, while decreasing $h$ results in the decrease of the error, indicating that the accuracy of the approximation improves.

In the context of training Neural ODEs, the loss function is typically defined as the error between the predicted trajectory and the ground-truth trajectory over a set of observed data points. In this work we consider the Absolute Error (AE),

$$l(\boldsymbol{\theta}) = \frac{1}{M} \sum_{m=1}^{M} |\hat{\boldsymbol{y}}_m(\boldsymbol{\theta}) - \boldsymbol{y}_m|.$$

However, since predictions $\hat{\boldsymbol{y}}_m(\boldsymbol{\theta})$ are computed using the contributions of a numerical method, equation (1), they are affected by $E_{\text{global}}^m(h, p)$ resulting in,

$$\hat{\boldsymbol{y}}_m(\boldsymbol{\theta}) = \hat{\boldsymbol{y*}}_m(\boldsymbol{\theta}) + E_{\text{global}}^m(h, p),$$

where $\hat{\boldsymbol{y*}}_m(\boldsymbol{\theta})$ is the Neural ODE prediction without the approximation error of the numerical method.

By incorporating the accumulated error into the predictions, the loss function is then given by,

$$l(\boldsymbol{\theta}) = \frac{1}{M} \sum_{m=1}^{M} |(\hat{\boldsymbol{y}*}_m(\boldsymbol{\theta}) + E_{\text{global}}^m(h, p)) - \boldsymbol{y}_m|.$$

Consequently, the accumulated error influences the computation of the loss function by introducing a term dependent on the numerical method into the predicted trajectory, thereby inducing variance in the loss. This variance impacts the optimisation process during training, thereby influencing parameter adjustments and rendering them dependent on the chosen numerical method at the training stage. As a result, employing different numerical methods for making predictions introduces discrepancies between the predicted trajectory and the trajectory learnt during training. Thus, predictions may deviate from anticipated trajectories based on training data, potentially leading to inaccuracies in model performance evaluation and decision-making.

### 3.2 EXPERIMENTAL RESULTS

To validate our theoretical analysis, we conduct preliminary numerical experiments focusing on a simple regression task involving the dynamics of a spiral ODE, as detailed by Chen et al. (2018) (Chen, 2018). Our aim is to demonstrate the influence of numerical methods even in simple tasks with shallow Neural ODEs and low training times [1].

We train a Neural ODE using a specific numerical method and evaluate its performance on both the training dataset (reconstruction) and a testing dataset (extrapolation) using the same numerical method employed during training. We compare this performance against that achieved with different numerical methods. Performance is evaluated by training two models with each method and computing the average AE and respective standard deviation (std).

The results for reconstruction and extrapolation are summarised in Appendix A Table 1 and Table 2, respectively. Notably, the numerical methods rk4, implicit_adams, dopri8, dopri5, bosh3, fehlberg2 and adaptive_heun exhibit consistent performance, suggesting that these methods could be used interchangeably for both training and testing without significant performance degradation in reproducing or extrapolating the learnt trajectory. Conversely, euler and explicit_adams demonstrate high performance variance, while also failing to learn the dynamics of the spiral ODE being evident from the high error values.

While this study explores the impact of numerical methods on Neural ODE performance using simplified conditions, further investigation is needed. Theoretical considerations suggest that more differences in numerical method orders may lead to performance divergence in complex scenarios. Thus, additional experiments with varied conditions, including complex datasets and deeper architectures, are essential to fully understand, and experimentally study, the behaviour of different numerical methods.

## 4   CONCLUSION

In this work, we show, from a theoretical analysis, that during the training of Neural ODEs, errors introduced by the chosen numerical method propagate through the optimisation process. The learnt parameters $\boldsymbol{\theta}$ are optimised to minimise a loss function $l(\boldsymbol{\theta})$, accounting for the errors inherent in the numerical method's approximation. Thus, when making predictions using a different numerical method, the discrepancies in errors (arising from differences in order or step size) lead to divergent approximations of the system dynamics. Consequently, predictions made with different numerical methods deviate from those obtained during training, resulting in higher prediction errors, especially if the model was trained for many iterations in which the method-induced errors accumulate significantly. Hence, careful consideration is imperative to ensure consistency across model training and prediction phases. The preliminary experimental results corroborate the theoretical insights however further numerical experiments are essential to evaluate the performance divergence of numerical methods in complex scenarios, considering factors such as dataset complexity, network depth, and training iterations.

---

[1]This does not encompass experiments on using the same numerical method with different step sizes.

ACKNOWLEDGMENTS

The authors acknowledge the funding by Fundação para a Ciência e Tecnologia (Portuguese Foundation for Science and Technology) through CMAT projects UIDB/00013/2020 and UIDP/00013/2020 and the funding by FCT and Google Cloud partnership through projects CPCA-IAC/AV/589164/2023 and CPCA-IAC/AF/589140/2023.

C. Coelho would like to thank FCT for the funding through the scholarship with reference 2021.05201.BD.

This work is also financially supported by national funds through the FCT/MCTES (PIDDAC), under the project 2022.06672.PTDC - iMAD - Improving the Modelling of Anomalous Diffusion and Viscoelasticity: solutions to industrial problems.

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

## A    NUMERICAL RESULTS

Table 1: Numerical results for the spiral ODE at reconstruction (average AE $\pm$ std).

| Training | | Fixed-step | | | | | Adaptive-step | | | | |
|---|---|---|---|---|---|---|---|---|---|---|---|
| Predictions | | euler | midpoint | rk4 | explicit_adams | implicit_adams | dopri8 | dopri5 | bosh3 | fehlberg2 | adaptive_heun |
| Fixed-step | euler | 5.4e-01 ± 2.3e-01 | 5.5e-01 ± 2.7e-02 | 5.6e-01 ± 2.6e-02 | 6.4e+01 ± 3.8e-02 | 5.6e-01 ± 2.6e-02 | 5.6e-01 ± 2.6e-02 | 5.6e-01 ± 2.6e-02 | 5.6e-01 ± 2.6e-02 | 5.6e-01 ± 2.6e-02 | 5.6e-01 ± 2.6e-02 |
| | midpoint | 1.9e+00 ± 9.6e-02 | 2.3e-01 ± 1.2e-01 | 2.3e-01 ± 1.2e-01 | 6.4e+01 ± 4.0e-01 | 2.3e-01 ± 1.2e-01 | 2.3e-01 ± 1.2e-01 | 2.3e-01 ± 1.2e-01 | 2.3e-01 ± 1.2e-01 | 2.3e-01 ± 1.1e-01 | 2.3e-01 ± 1.2e-01 |
| | rk4 | 2.6e+00 ± 3.0e-01 | 4.0e-01 ± 3.0e-01 | 4.0e-01 ± 3.0e-01 | 6.4e+01 ± 2.0e-01 | 4.0e-01 ± 3.0e-01 | 4.0e-01 ± 3.0e-01 | 4.0e-01 ± 3.0e-01 | 4.0e-01 ± 3.0e-01 | 4.0e-01 ± 3.0e-01 | 4.0e-01 ± 3.0e-01 |
| | explicit_adams | 2.0e+00 ± 3.2e-01 | 2.9e-01 ± 5.5e-02 | 2.9e-01 ± 5.8e-02 | 6.4e+01 ± 6.1e-0 | 2.9e-01 ± 5.8e-02 | 2.9e-01 ± 5.8e-02 | 2.9e-01 ± 5.8e-02 | 2.9e-01 ± 5.8e-02 | 2.9e-01 ± 5.8e-02 | 2.9e-01 ± 5.8e-0 |
| | implicit_adams | 2.8e+00 ± 1.5e-01 | 4.8e-01 ± 2.4e-01 | 4.7e-01 ± 2.5e-01 | 6.3e+01 ± 2.9e-02 | 4.7e-01 ± 2.5e-01 | 4.7e-01 ± 2.5e-01 | 4.7e-01 ± 2.5e-01 | 4.7e-01 ± 2.5e-01 | 4.7e-01 ± 2.5e-01 | 4.7e-01 ± 2.5e-01 |
| Adaptive-step | dopri8 | 1.9e+00 ± 3.0e-02 | 6.6e-01 ± 6.5e-02 | 6.6e-01 ± 6.6e-02 | 6.3e+01 ± 2.3e-01 | 6.6e-01 ± 6.6e-02 | 6.6e-01 ± 6.6e-02 | 6.6e-01 ± 6.6e-02 | 6.6e-01 ± 6.6e-02 | 6.6e-01 ± 6.6e-02 | 6.6e-01 ± 6.6e-02 |
| | dopri5 | 2.2e+00 ± 2.0e-01 | 3.8e-01 ± 2.1e-01 | 3.8e-01 ± 2.0e-01 | 6.4e+01 ± 1.0e+00 | 3.8e-01 ± 2.0e-01 | 3.8e-01 ± 2.0e-01 | 3.8e-01 ± 2.0e-01 | 3.8e-01 ± 2.0e-01 | 3.8e-01 ± 2.0e-01 | 3.8e-01 ± 2.0e-01 |
| | bosh3 | 1.9e+00 ± 2.0e-01 | 2.0e-01 ± 1.6e-02 | 2.0e-01 ± 1.3e-02 | 6.4e+01 ± 2.4e-01 | 2.0e-01 ± 1.3e-02 | 2.0e-01 ± 1.3e-02 | 2.0e-01 ± 1.3e-02 | 2.0e-01 ± 1.3e-02 | 2.0e-01 ± 1.3e-02 | 2.0e-01 ± 1.3e-02 |
| | fehlberg2 | 2.3e+00 ± 2.5e-01 | 5.2e-01 ± 1.6e-01 | 5.1e-01 ± 1.5e-01 | 6.4e+01 ± 1.5e-01 | 5.1e-01 ± 1.5e-01 | 5.1e-01 ± 1.5e-01 | 5.1e-01 ± 1.5e-01 | 5.1e-01 ± 1.5e-01 | 5.1e-01 ± 1.5e-01 | 5.1e-01 ± 1.5e-01 |
| | adaptive_heun | 2.0e+00 ± 2.0e-01 | 6.1e-01 ± 3.6e-02 | 6.1e-01 ± 3.8e-02 | 6.3e+01 ± 9.0e-01 | 6.1e-01 ± 3.8e-02 | 6.1e-01 ± 3.8e-02 | 6.1e-01 ± 3.8e-02 | 6.1e-01 ± 3.8e-02 | 6.1e-01 ± 3.8e-02 | 6.1e-01 ± 3.8e-02 |

Table 2: Numerical results for the spiral ODE at extrapolation (average AE $\pm$ std).

| Training | | Fixed-step | | | | | Adaptive-step | | | | |
|---|---|---|---|---|---|---|---|---|---|---|---|
| Predictions | | euler | midpoint | rk4 | explicit_adams | implicit_adams | dopri8 | dopri5 | bosh3 | fehlberg2 | adaptive_heun |
| Fixed-step | euler | 2.4e+00 ± 2.8e-01 | 5.5e-01 ± 2.7e-02 | 5.5e-01 ± 2.6e-02 | 1.3e+02 ± 4.7e-02 | 5.5e-01 ± 2.6e-02 | 5.5e-01 ± 2.6e-02 | 5.5e-01 ± 2.6e-02 | 5.5e-01 ± 2.6e-02 | 5.5e-01 ± 2.6e-02 | 5.5e-01 ± 2.6e-02 |
| | midpoint | 3.5e+00 ± 2.8e-01 | 2.4e-01 ± 1.9e-01 | 2.3e-01 ± 1.1e-01 | 1.3e+02 ± 6.1e-01 | 2.3e-01 ± 1.3e-01 | 2.3e-01 ± 1.2e-0 | 2.3e-01 ± 1.2e-0 | 2.3e-01 ± 1.2e-01 | 2.3e-01 ± 1.1e-01 | 2.3e-01 ± 1.2e-01 |
| | rk4 | 4.6e+00 ± 3.9e-01 | 4.0e-01 ± 3.5e-01 | 4.0e-01 ± 3.0e-01 | 1.3e+02 ± 2.6e-0 | 4.0e-01 ± 3.1e-01 | 4.0e-01 ± 3.0e-01 | 4.0e-01 ± 3.0e-0 | 4.0e-01 ± 3.0e-01 | 4.0e-01 ± 3.0e-01 | 4.0e-01 ± 3.0e-01 |
| | explicit_adams | 3.8e+00 ± 6.3e-01 | 2.9e-01 ± 2.5e-02 | 2.9e-01 ± 6.0e-02 | 1.3e+02 ± 4.5e-01 | 2.9e-01 ± 4.2e-02 | 2.9e-01 ± 5.8e-02 | 2.9e-01 ± 5.8e-02 | 2.9e-01 ± 5.8e-02 | 2.9e-01 ± 5.8e-02 | 2.9e-01 ± 5.8e-02 |
| | implicit_adams | 4.9e+00 ± 1.1e-01 | 5.1e-01 ± 1.8e-01 | 4.7e-01 ± 2.5e-01 | 1.3e+02 ± 3.8e-02 | 4.8e-01 ± 2.4e-01 | 4.7e-01 ± 2.5e-01 | 4.7e-01 ± 2.5e-01 | 4.7e-01 ± 2.5e-01 | 4.7e-01 ± 2.5e-0 | 4.7e-01 ± 2.5e-01 |
| Adaptive-step | dopri8 | 3.7e+00 ± 1.4e-01 | 7.0e-01 ± 3.4e-02 | 6.6e-01 ± 6.6e-02 | 1.3e+02 ± 6.7e-01 | 6.7e-01 ± 6.0e-02 | 6.6e-01 ± 6.6e-02 | 6.6e-01 ± 6.6e-02 | 6.6e-01 ± 6.6e-02 | 6.6e-01 ± 6.6e-02 | 6.6e-01 ± 6.6e-02 |
| | dopri5 | 4.1e+00 ± 2.9e-01 | 3.8e-01 ± 2.9e-01 | 3.8e-01 ± 2.0e-01 | 1.3e+02 ± 2.0e+00 | 3.8e-01 ± 2.2e-01 | 3.8e-01 ± 2.0e-01 | 3.8e-01 ± 2.0e-01 | 3.8e-01 ± 2.0e-01 | 3.8e-01 ± 2.0e-01 | 3.8e-01 ± 2.0e-01 |
| | bosh3 | 3.4e+00 ± 2.7e-01 | 2.0e-01 ± 9.8e-02 | 2.0e-01 ± 1.1e-01 | 1.3e+02 ± 7.7e-01 | 2.0e-01 ± 2.9e-02 | 2.0e-01 ± 1.3e-02 | 2.0e-01 ± 1.3e-02 | 2.0e-01 ± 1.3e-02 | 2.0e-01 ± 1.3e-02 | 2.0e-01 ± 1.3e-02 |
| | fehlberg2 | 4.2e+00 ± 5.2e-01 | 5.9e-01 ± 1.4e-01 | 5.1e-01 ± 1.5e-01 | 1.3e+02 ± 1.1e-02 | 5.3e-01 ± 1.5e-01 | 5.1e-01 ± 1.5e-01 | 5.1e-01 ± 1.5e-01 | 5.1e-01 ± 1.5e-01 | 5.1e-01 ± 1.5e-01 | 5.1e-01 ± 1.5e-01 |
| | adaptive_heun | 3.8e+00 ± 8.4e-01 | 6.3e-01 ± 2.5e-02 | 6.1e-01 ± 3.9e-02 | 1.3e+02 ± 1.9e+00 | 6.2e-01 ± 2.7e-02 | 6.1e-01 ± 3.8e-02 | 6.1e-01 ± 3.8e-02 | 6.1e-01 ± 3.8e-02 | 6.1e-01 ± 3.8e-02 | 6.1e-01 ± 3.8e-02 |

