# OpenReview forum: "Consistency Matters: Neural ODE Parameters are Dependent on the Training Numerical Method"
_ICLR.cc/2024/Workshop/AI4DiffEqtnsInSci — AI4DiffEqtnsInSci @ ICLR 2024 Poster_

### Official Review · Reviewer_dBUN · 2024-02-14
**Lack of solid contribution and conclusive results**

**Rating:** 4
**Confidence:** 5

**Review:**

The concept being discussed in the paper regarding effect of the numerical method for solving the forward problem affecting the inverse problem is not novel and a widely known numerical analysis fact. The authors do not contribute significant experimentation to justify that they furthered the understanding of this phenomena.

---

### Official Review · Reviewer_ktU9 · 2024-02-26
**Interesting if theory holds**

**Rating:** 7
**Confidence:** 3

**Review:**

The authors propose an interesting result in regards to neural ODEs. The authors claim that training and testing of neural ODEs should be performed by the same ODE solver to obtain best results for in and out-of-distribution tasks.

I cannot check the validity of the theoretical proof. In case, the  proof holds perfectly - the paper must be accepted for the workshop. In case there are nuances in the theoretical proof, they must be mentioned in the paper for clarity. Overall the idea of the paper is relevant in the context of the workshop. For authors future work, the following points could be interesting.

In the first place, why would anyone try different ODE solvers for training and testing. Is it not intuitive to use the same ODE solver?
There are several neural ODE based methods, how can the results of the proposed work be generalized?

---

### Meta-Review · Area_Chair_2Fs7 · 2024-02-26

**Recommendation:** Accept (Poster)

**Metareview:**

The authors look at an important problem of the effect of numerical discretization in Neural ODEs, which has been studied in the literature as Reviewer dBUN mentioned. In particular, several key references are missing, including Onken and Ruthotto, "Discretize-Optimize vs. Optimize-Discretize for Time-Series Regression and Continuous Normalizing Flows", 2020 and Krishnapriyan et al., "Learning continuous models for continuous physics", 2022. I vote for acceptance pending that the authors add these and other references from the literature and further clarify their contribution.

---

### Decision · Program_Chairs · 2024-02-28

Accept (Poster)